# Parallel evolution of influenza across multiple spatiotemporal scales

Katherine S Xue[1,2], Terry Stevens-Ayers[3], Angela P Campbell[3†], Janet A Englund[4,5], Steven A Pergam[3,6,7], Michael Boeckh[3,6,7], Jesse D Bloom[1,2]*

[1]Department of Genome Sciences, University of Washington, Seattle, United States; [2]Basic Sciences Division and Computational Biology Program, Fred Hutchinson Cancer Research Center, Seattle, United States; [3]Vaccine and Infectious Disease Division, Fred Hutchinson Cancer Research Center, Seattle, United States; [4]Seattle Children's Research Institute, Seattle, United States; [5]Department of Pediatrics, University of Washington, Seattle, United States; [6]Department of Medicine, University of Washington, Seattle, United States; [7]Clinical Research Division, Fred Hutchinson Cancer Research Center, Seattle, United States

**Abstract** Viral variants that arise in the global influenza population begin as *de novo* mutations in single infected hosts, but the evolutionary dynamics that transform within-host variation to global genetic diversity are poorly understood. Here, we demonstrate that influenza evolution within infected humans recapitulates many evolutionary dynamics observed at the global scale. We deep-sequence longitudinal samples from four immunocompromised patients with long-term H3N2 influenza infections. We find parallel evolution across three scales: within individual patients, in different patients in our study, and in the global influenza population. In hemagglutinin, a small set of mutations arises independently in multiple patients. These same mutations emerge repeatedly within single patients and compete with one another, providing a vivid clinical example of clonal interference. Many of these recurrent within-host mutations also reach a high global frequency in the decade following the patient infections. Our results demonstrate surprising concordance in evolutionary dynamics across multiple spatiotemporal scales.

*For correspondence: jbloom@fredhutch.org

Present address: †Centers for Disease Control and Prevention, Atlanta, Georgia

## Introduction

Viruses rapidly acquire *de novo* mutations as they replicate within infected hosts (*Andino and Domingo, 2015*), but only a small fraction of these variants transmit between hosts and eventually fix on a global scale. Within hosts, a mutation's impact on viral replication and immunogenicity affect whether it increases in frequency. At larger scales of space and time, transmission bottlenecks (*Varble et al., 2014*; *Poon et al., 2016*) and host heterogeneity also shape viral genetic diversity. The selective pressures at these various scales reflect complex molecular, immunological, and epidemiological constraints (*Grenfell et al., 2004*; *Pybus and Rambaut, 2009*; *Luksza and Lässig, 2014*; *Neher et al., 2016*), which have formed the basis of recent efforts to forecast influenza evolution (*Luksza and Lässig, 2014*; *Neher et al., 2016*, *Neher et al., 2014*; *Lässig et al., 2017*).

Influenza's rapid global evolution has been the subject of intense study (*Ghedin et al., 2005*; *Rambaut et al., 2008*), but the origins of this variation within single infected hosts are still poorly understood. Recent deep-sequencing studies of human clinical samples suggest that influenza accumulates relatively limited genetic diversity within hosts during most acute infections (*Dinis et al., 2016*; *Poon et al., 2016*; *Sobel Leonard et al., 2016*; *Debbink et al., 2017*), in line with earlier studies in dogs and horses (*Murcia et al., 2010*; *Hoelzer et al., 2010*). Some within-host mutations may confer novel antigenic properties (*Dinis et al., 2016*), but most lack clear functional

**eLife digest** Influenza or flu viruses change fast to escape the body's defenses. While a single course of vaccines will protect someone against polio or measles for their whole life, people need a new flu shot every year to be protected against influenza. Also, some years the flu vaccine is not as effective as hoped because the virus has changed in an unpredictable way.

All of the change that happens in flu viruses around the world ultimately begins in individual infections, as random mistakes or mutations in the virus's genetic material that arise as the viruses replicate. Mutations that help the virus aid in its ability to spread from person to person, and eventually spread around the world. As such, understanding flu's evolution within individual people may help scientists to understand and eventually predict how it changes worldwide. Yet, unlike some viral infections that last months or years, flu infections are usually short and over in a few days. This makes it harder to measure how the viruses change over time in a single infection.

To get around this issue, Xue et al. analyzed flu samples taken over several weeks from four cancer patients who had longer-than-average flu infections because of their weaker immune systems. In some cases, the exact same mutations were seen in viruses from two or more of the patients. Also, some of the mutations that happened within the patients were the same mutations that later went on to spread around the world. These findings show that the flu virus can change in a single person in some of the same ways that it has been seen to change around the world.

Xue et al. studied how flu changes in people with weak immune systems, who are infected for longer periods of time. Further studies are needed to reveal more about how flu viruses evolve in the more typical, shorter infections in otherwise healthy people. This kind of investigation is becoming easier because new methods are making it possible to examine the genetic material from many viruses at once. It is hoped that eventually, detecting mutations in individual infections could help predict how viruses will change worldwide, which might help researchers to design vaccines that will be more effective each year.

interpretation. Altogether, it remains unclear how influenza's within-host diversity is transformed into global evolution.

Influenza infections usually last less than a week and provide limited opportunity for longitudinal study. But among some immunocompromised patients, infections can last weeks or months (*Nichols et al., 2004*; *Memoli et al., 2014*), making it possible to examine longer-term within-host evolutionary dynamics (*Rocha et al., 1991*; *McMinn et al., 1999*; *Rogers et al., 2015*). Here, we use deep-sequencing to characterize the evolutionary dynamics of influenza within immunocompromised hosts. We identify a small set of mutations that arise repeatedly within individual patients, across multiple patients in our study, and at the global scale, revealing surprising similarities in evolutionary dynamics across multiple spatiotemporal scales.

## Results

### The same mutations often arise in multiple patients

We deep-sequenced 37 viral samples collected longitudinally from four immunocompromised patients with long-term H3N2 influenza infections in the 2005–2006 and 2006–2007 seasons (*Figure 1*). These patients developed influenza infections in the months after receiving hematopoietic cell transplantations when immune cell counts were still low, and nasal wash samples were collected approximately every week. All patients were treated with the neuraminidase inhibitor oseltamivir for at least some duration of their infections (*Campbell et al., 2015*) (*Figure 1*, *Figure 1—figure supplement 1*).

We sequenced the full viral genome to high coverage directly from patient nasal wash samples by using influenza-specific reverse transcription and PCR (*Hoffmann et al., 2001*) to enrich for viral genetic material (*Figure 2—figure supplement 1*). To limit the impact of library preparation and sequencing errors on estimates of variant frequency (*McCrone and Lauring, 2016*), we prepared sequencing libraries in duplicate for each sample, beginning from separate reverse-transcription

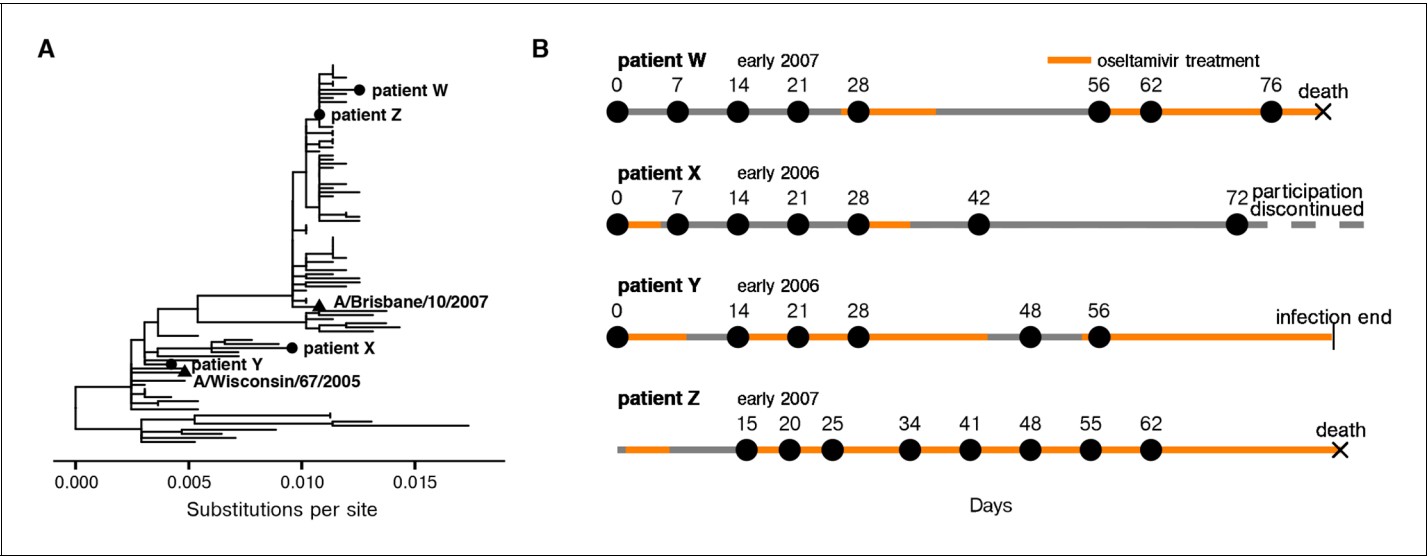

**Figure 1.** Long-term H3N2 influenza infections in four immunocompromised patients. (**A**) Phylogenetic relationship between initial patient consensus sequences and 63 unique circulating influenza strains collected in the USA from 2004 to 2007, as inferred from the HA gene. (**B**) Overview of patient influenza infections and treatments. Periods of oseltamivir treatment are shown in orange. Dates of sequenced nasal wash samples are calculated relative to the first influenza-positive nasal wash. Low-quality samples are not shown here and were excluded from downstream analysis. Materials and methods and *Figure 1—figure supplement 1* give full clinical histories.

The following figure supplement is available for figure 1:

**Figure supplement 1.** Summary of patient infections.

reactions. We excluded from downstream analyses eight low-quality samples for which sequencing coverage was low or variant frequencies differed greatly between replicates (*Figure 2—figure supplement 1*).

Across the influenza genome, *de novo* mutations arise most commonly in the surface proteins hemagglutinin (HA) and neuraminidase (NA) (*Figure 2A*), which undergo rapid global evolution (*Bhatt et al., 2011*). These mutations fluctuate in frequency but rarely fix, showing that complex evolutionary dynamics can emerge within single infected individuals (*Figure 2B*, *Figure 2—figure supplements 2–5*). We focused on within-host mutations that reached a frequency of at least 5% in two independent sequencing replicates from any patient sample. Many nonsynonymous mutations occur at sites that affect the antigenicity of HA (*Koel et al., 2013*) and the antiviral sensitivity of NA (*Baz et al., 2006*; *van der Vries et al., 2013*) (*Figure 2—figure supplement 6*). In NA in particular, we observe the emergence and persistence of mutations T242I and R292K, which are known to be associated with oseltamivir resistance (*Baz et al., 2006*; *van der Vries et al., 2013*), a phenomenon of strong clinical importance (*Renaud et al., 2011*) (*Figure 2—figure supplements 2–5*).

In several cases, the same mutations arise independently and reach high frequency in multiple patients (*Figure 2C*). We identified nine sites in the influenza genome where parallel mutations arose in two or more patients in our study: five in HA, three in NA, and one in the nonstructural (NS) segment (*Figure 2C*, *Figure 2—figure supplement 7*; HA: p<0.001; NA: p<0.01; permutation test). In subsequent analyses, we focused primarily on HA because of its prominent role in antigenic evolution (*Koel et al., 2013*).

## Recurrent mutations drive clonal interference within individual patients

Although the same HA mutations arise in multiple patients, we found that evolutionary outcomes sometimes diverge. For instance, A138S arises in patients W and Z, but it fixes only in patient Z. In three patients, N225D reaches a detectable frequency, but it fixes only in patient X (*Figure 2C*).

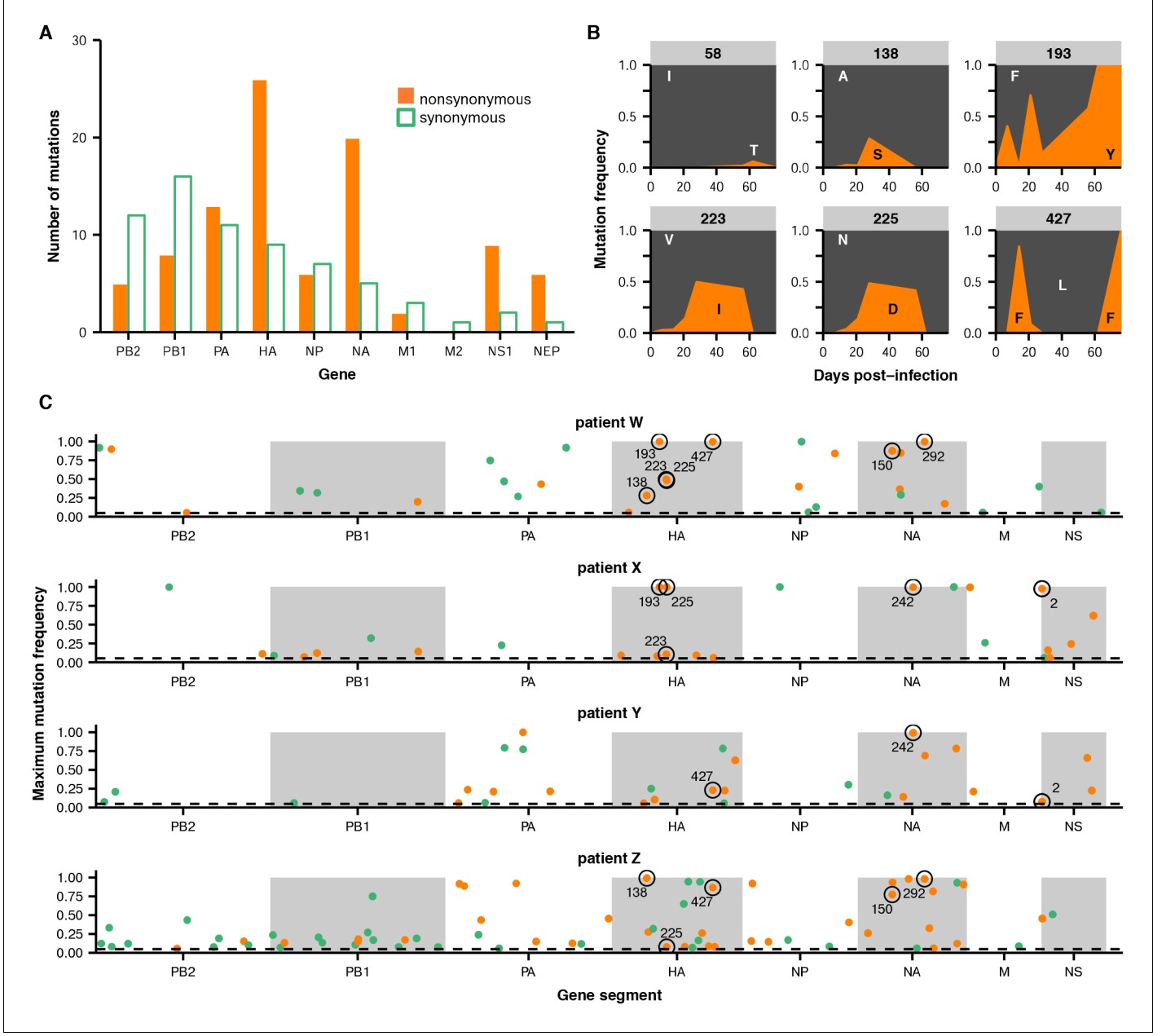

**Figure 2.** Within-host influenza variants. (**A**) Number of nonsynonymous (orange) and synonymous (green) variants in each influenza gene. We identified within-host viral mutations that reached a frequency of at least 5% in two independent sequencing replicates from any patient sample. (**B**) Frequencies over time for all HA mutations in patient W. Each subplot represents a site in HA and is labeled by codon number. Ancestral identities are colored in gray and mutant ones in orange. (**C**) Maximum frequencies reached by all nonsynonymous (orange) and synonymous (green) mutations in each patient. Mutations circled in black emerged independently in multiple patients and are labeled by codon number. The dotted line indicates the minimum frequency threshold of 5%. Materials and methods and *Figure 2—figure supplement 1* describe procedures used for variant calling and quality control. *Figure 2—figure supplements 2–5* give full frequency trajectories for all mutations in all patients. *Figure 2—figure supplement 6* shows mutations in HA and NA on their respective crystal structures. *Figure 2—figure supplement 7* describes permutation tests that assess the significance of the observed parallelism between patients.

The following source data and figure supplements are available for figure 2:

**Source data 1.** Primers used for viral deep sequencing.

**Figure supplement 1.** Sample quality controls.

*Figure 2 continued on next page*

*Figure 2 continued*

**Figure supplement 2.** Within-host variants in patient W.
**Figure supplement 3.** Within-host variants in patient X.
**Figure supplement 4.** Within-host variants in patient Y.
**Figure supplement 5.** Within-host variants in patient Z.
**Figure supplement 6.** Sites of within-host mutation.
**Figure supplement 7.** Permutation tests for parallel evolution between patients.

We suspected that the complex dynamics of these within-host mutations might arise from competition among mutant lineages. The influenza genome consists of eight linear segments that freely reassort with one another but do not recombine (*Boni et al., 2008*), meaning that each segment evolves clonally. In the absence of homologous recombination, lineages carrying beneficial mutations rise and fall in frequency as they compete with one another, making it harder for any one variant to fix. This phenomenon, known as clonal interference, has been characterized extensively in experimental evolution (*Hegreness et al., 2006*; *Kao and Sherlock, 2008*; *Lang et al., 2013*; *Neher, 2013*) and affects influenza's global evolution (*Strelkowa and Lässig, 2012*).

We examined clonal dynamics within individual patients by analyzing patterns of linkage among within-host mutations. We identified read pairs that spanned multiple variable sites to infer linkage, and we summarized these relationships as haplotypes: for instance, '0000' represents ancestral residues at four variable sites, and '1100' represents a double-mutant at the first two sites (*Figure 3A*).

In several instances, the same mutations arise in parallel on distinct genetic backgrounds within the same patient—echoing our observation that these same mutations arise in parallel in multiple patients in our study. In patient X, lineages carrying S193Y and N225D initially compete, but a double-mutant carrying both mutations eventually fixes (*Figure 3B*). The A138S and F193Y mutations also arise multiple times in parallel in patient W: once on the ancestral haplotype '0000' to form the single-mutant '1000' and '0100' lineages; once on these single-mutant lineages to form the double-mutant '1100'; and once on the double-mutant '0011' to form the triple-mutant '1011' and '0111' lineages (*Figure 3C*). These recurrent mutations also contribute to the large number of clonal lineages present. Several weeks into patient W's infection, we observe at least five distinct HA lineages at a frequency of at least 5%, and the lineages differ from each other by one to three nonsynonymous mutations (*Figure 3C*). Eventually, all lineages that carry A138S, V223I, and N225D are outcompeted by a lineage that carries F193Y.

Our analysis shows that in large, clonally evolving influenza populations within hosts, a small set of beneficial mutations repeatedly arise and compete against one another in various combinations. Although many of these beneficial mutations are selected in parallel in multiple patients, the unpredictability of clonal competition determines which mutations eventually fix.

## Within-host variants often arise at sites that are polymorphic in influenza globally

We compared viral mutations that arose within our patients and at the global scale. Strikingly, many of the HA mutations that arise in parallel in multiple patients in our study also reach a high global frequency, which may reflect concordant antigenic selection at the within-host and global scales. We identified all variants that reached a frequency of at least 10% in any given year after 2000 in the GISAID database of global influenza sequences (*Bogner et al., 2006*) and compared them to variants that we identified in the patients in our study.

In HA, most sites that varied within hosts also varied in the global influenza population, compared to about a quarter of such sites in the other influenza genes (*Figure 4*). We tested whether this overlap between sites of variation within patients and globally was greater than expected by chance for HA, NA, and the rest of the viral genome combined. We calculated the expected overlap when the

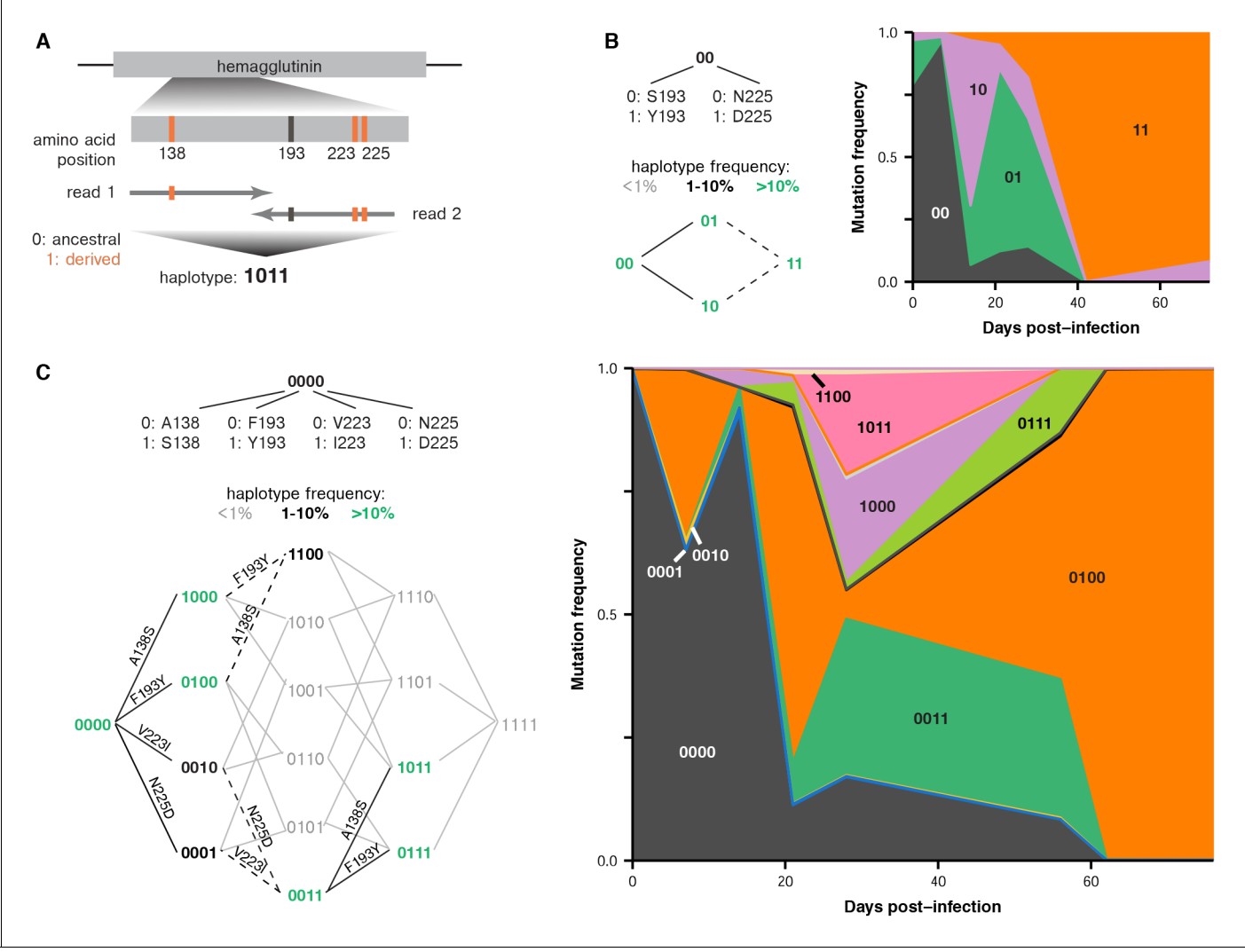

**Figure 3.** Parallel emergence of the same mutations within single infected hosts. (A) Method for inferring partial haplotypes from short-read sequencing data. We identified paired-end reads that spanned multiple sites of interest along a gene and determined whether the read carried the ancestral or derived allele at each site. (B) Frequencies of haplotypes at HA sites 193 and 225 in patient X. Evolutionary paths from the ancestral to double-mutant state are shown, with haplotypes colored according to their maximum frequency during the infection. Solid black lines connect pairs of haplotypes that are both present at a frequency of above 1% and that unambiguously occurred through the indicated mutation. Dashed lines indicate that multiple mutations could have produced a particular haplotype. Gray lines indicate that a mutation did not arise at a detectable frequency on a particular haplotype background. (C) Frequencies of haplotypes at HA sites 138, 193, 223, and 225 in patient W. *Figure 3—figure supplement 1* estimates the rate of PCR recombination as described in Materials and methods. *Figure 3—figure supplement 2* and *3* show the number of paired-end reads that spanned the mutations in the haplotypes in patients X and W.

The following figure supplements are available for figure 3:

**Figure supplement 1.** Estimate of PCR recombination rate.

**Figure supplement 2.** Number of paired-end reads used to infer haplotype dynamics in patient X.

**Figure supplement 3.** Number of paired-end reads used to infer haplotype dynamics in patient W.

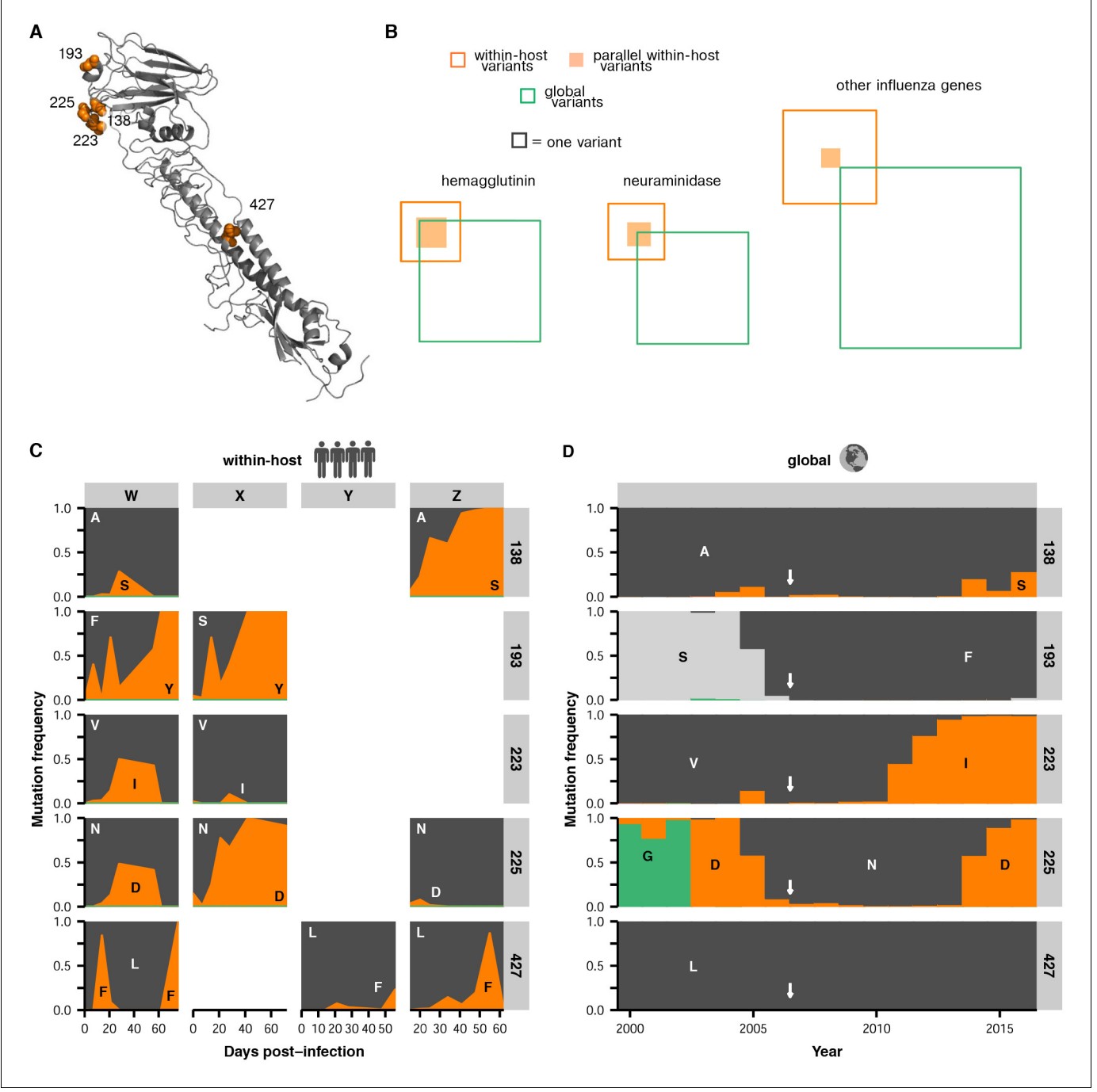

**Figure 4.** Parallel mutations at within-host and global scales. (**A**) Sites of parallel within-host mutation plotted on an HA crystal structure (PDB 4HMG [*Weis et al., 1990*]). (**B**) Overlap of within-host (orange) and global (green) variable sites in HA, NA, and all other influenza genes. Sites at which mutations arise in more than one patient are indicated in solid orange. We defined global variable sites as those at which a variant reached a frequency of at least 10% in a given year after 2000 in the GISAID database of global influenza sequences (*Bogner et al., 2006*). Numbers of within-host and global mutations are given in *Figure 4—source data 1*. (**C**) Mutation frequencies over time within individual patients for parallel within-host mutations in HA. Ancestral identities are colored in gray and mutant ones in orange. (**D**) Global variant frequencies between 2000 and 2015 in H3N2 influenza at sites of parallel within-host mutation in HA. The approximate timing of the patient infections (2006–2007) is indicated by a white arrow. *Figure 4—figure supplement 1* displays variant frequencies for all sites of parallel mutation at the within-host and global scales. *Figure 4—figure supplement 2* describes permutation tests that assess the significance of the overlap in mutations at the within-host and global scales.

The following source data and figure supplements are available for figure 4:

**Source data 1.** Overlap of mutations at the within-host and global scales.

*Figure 4 continued on next page*

*Figure 4 continued*

**Figure supplement 1.** Parallel mutations at within-host and global scales.

**Figure supplement 2.** Permutation tests for parallel evolution across within-host and global scales.

observed number of within-host and global variants were drawn at random from each gene (*Figure 4—figure supplements 1–2*). Not all sites are expected to tolerate mutation, so we also performed simulations where we only considered sites for which there was variation in human H3N2 influenza globally between 2000 and 2015: for instance, in HA about 25% of codon sites show no variation within the GISAID database. We found significant parallelism in HA ($p<0.01$), but not in NA or in the rest of the genome ($p>0.05$) when we consider all sites of global variation. This parallelism in HA evolution remains statistically significant at a 0.05 threshold until we assume that less than 50% of HA codon sites tolerate variation.

The parallelism is especially striking at the sites of HA mutations found in multiple patients in our study. In particular, four of the five sites of recurrent within-host mutation in HA are also sites of global influenza variation (*Figure 4*, *Figure 4—source data 1*). The V223I and N225D mutations arise in multiple patients, and then fix globally in the decade after the patient infections (*Figure 4D*). Mutations also reach high global frequencies at sites 138 and 193, although the F193 and S193 variants that spread globally differ from the Y193 variant that arises within our patients. However, the concordance is incomplete. Mutation L427F reaches a frequency of >75% in three patients but is rare or nonexistent in influenza globally (*Figure 4D*), suggesting that this mutation may have within-host benefits that are not reflected in global evolution. But overall across hemagglutinin, within-host variants tend to arise at sites that vary on the global scale.

## Discussion

It is remarkable that influenza evolution shows such extensive parallelism at these disparate spatio-temporal scales despite heterogeneity in host immunity, viral genetic background, and the severity and duration of infection. In particular, the immunocompromised patients in our study had complex underlying conditions and diverse immune histories. Notably, the four HA sites that displayed parallel within-host and global evolution in our study (138, 193, 223, and 225) also gave rise to mutations in another study that used Sanger sequencing to analyze laboratory-passaged influenza isolated longitudinally from an immunocompromised child (*Baz et al., 2006*). Another previous study used hemagglutination inhibition assays to show that antigenic drift of influenza within an immunocompromised patient resembled global antigenic change (*McMinn et al., 1999*). These similarities further support our finding that influenza evolution shows parallelism across diverse patients.

The parallel evolution that we observe in influenza at the within-host and global scales contrasts with HIV, where similar mutations can arise within hosts that share an HLA type, but tend to revert upon transmission to recipients with different HLA types (*Leslie et al., 2004*; *Herbeck et al., 2006*; *Lemey et al., 2006*; *Zanini et al., 2015*). Part of the difference may be that immune epitopes in influenza are broadly similar among individuals, with some exceptions (*Li et al., 2013*; *Linderman et al., 2014*), whereas the targets of anti-HIV immunity vary more widely due to patient-specific factors like HLA type.

We suggest that parallelism in HA evolution may emerge from the confluence of several evolutionary conditions (*Lässig et al., 2017*). First, if selection acts concordantly across environments, it will favor a common set of beneficial mutations. Second, in a constrained evolutionary landscape, this set of beneficial mutations will be relatively small. Finally, given sufficiently large population sizes, high mutation rates, and time, these beneficial mutations will emerge and be selected to detectable frequencies. Our observation that similar mutations arise repeatedly within single patients, within multiple different patients, and at the global scale, suggests that at least some of these conditions may hold true.

The parallelism and extensive evolution that we observe in long-term influenza infections contrasts with the limited within-host variation found in prior studies, which sample from acute infections

of immunocompetent hosts (*Murcia et al., 2010*; *Hoelzer et al., 2010*; *Dinis et al., 2016*; *Debbink et al., 2017*; *Sobel Leonard et al., 2016*; *Poon et al., 2016*). For instance, one recent study deep-sequenced HA from several hundred patients but only found a small number of antigenic variants, and mostly at low frequencies (*Dinis et al., 2016*). But our study suggests that influenza may experience many of the same selective pressures within acute infections as it does globally, even if the short durations of these infections make it difficult for selected mutations to reach frequencies that are detectable with current methods. We suggest that within-host viral diversity may act as a noisy early measurement of global viral evolution, shaped by some of the same immunological and evolutionary constraints. As high-throughput sequencing continues to improve, detailed characterization of within-host variation will be increasingly valuable for understanding how molecular, immunological, and epidemiological forces interact to shape viral evolution.

## Materials and methods

### Patient material

Samples were prospectively collected during a surveillance study for respiratory viruses performed in allogeneic hematopoietic stem cell transplant (HCT) recipients undergoing transplantation between December 2005 and February 2010 at Fred Hutchinson Cancer Research Center (*Campbell et al., 2015*). Following written informed consent, weekly nasal wash samples (or nasopharyngeal swabs if nasal wash samples were precluded clinically) and oropharyngeal swab specimens were obtained at least once before and weekly after HCT up to 100 days. Afterwards, samples were collected as long as the patients continued to test positive for respiratory viruses, if they developed new symptoms, or at least every three months until one year post-transplantation. Nasal wash samples were collected using 5 mL of saline per nostril, and combined with oropharyngeal swabs for real-time PCR testing for a panel of 12 respiratory viruses, including influenza A and B. Samples were considered positive if the assay's cycle threshold was less than 40, for a limit of detection of approximately 2000 viral copies/mL. All samples sequenced in this study tested positive for influenza A. The timing of each sample during an infection was calculated as the number of days since the first influenza-positive nasal wash for that patient.

Descriptions of individual patients and their clinical courses are summarized below, with detailed information in *Figure 1—figure supplement 1*. All patients were severely immunocompromised: although their influenza infections occurred after transplant engraftment, their lymphocyte counts remained well below those found in immunocompetent individuals, and they were concurrently treated with immunosuppressive medications. Influenza sometimes co-occurred with other respiratory viruses, and the patients were frequently taking multiple antiviral and antibiotic medications at any given point in the infection.

### Patient W

A female in the 25–44 age group developed upper respiratory symptoms in early 2007, 30 days after receiving a non-myeloablative HCT for Hodgkin's disease and 18 days following engraftment. Patient nasal wash samples repeatedly tested positive for influenza A for the next 80 days until the patient died of pulmonary failure, with diffuse alveolar damage found on autopsy. The patient received a 12 day course of oseltamivir at 75 mg PO BID approximately 30 days into the infection and was treated continuously with oseltamivir for the last 26 days of her life, first at 75 mg PO BID and then increasing to 150 mg PO BID. The patient was co-infected with coronavirus for the duration of the influenza infection and also tested positive for human metapneumovirus for the last 26 days of her life.

### Patient X

A male in the 65+ age group developed upper respiratory symptoms in early 2006, 45 days after receiving a non-myeloablative HCT for Hodgkin's disease. Patient nasal wash samples repeatedly tested positive for influenza A for the next 72 days, after which the patient chose to discontinue study participation. The patient was treated with two courses of oseltamivir: a 5 day course at 75 mg PO BID following the first positive nasal wash, and an 8 day course at 75 mg PO BID approximately four weeks into the infection. The patient also tested positive for cytomegalovirus (CMV) and *Aspergillus* early in the influenza infection.

### Patient Y

A male in the 45–64 age group developed upper respiratory symptoms in spring 2006, 62 days after receiving a non-myeloablative HCT for acute myeloid leukemia (AML) and 52 days after engraftment. Patient nasal wash samples repeatedly tested positive for influenza A for the next 77 days, after which the patient began testing negative. The patient was treated with three courses of oseltamivir: an 8 day course following the first positive nasal wash, a 30 day course beginning approximately two weeks into the infection, and a second 30 day course starting approximately seven weeks into the infection, all at 75 mg PO BID. The patient also intermittently tested positive for CMV and coronavirus during the influenza infection.

### Patient Z

A male in the 65+ age group developed upper respiratory symptoms in early 2007, 197 days after receiving a non-myeloablative HCT for AML and 175 days after engraftment. Nasal wash samples repeatedly tested positive for influenza A over the next 69 days, after which monitoring ceased due to severe illness, and the patient died 15 days after the last influenza-positive sample from relapsed AML. The patient was treated with two courses of oseltamivir: a 6 day course at 150 mg PO BID following the first flu-positive nasal wash, and a 66 day course starting approximately two weeks into the infection that began at 150 mg PO QD and increased to 150 mg PO BID. The patient also received 30 g of IVIG 46 days into the flu infection. The patient intermittently tested positive for respiratory syncytial virus over the same period and also experienced Epstein-Barr viremia.

## Viral deep sequencing

To deep-sequence viral populations, we extracted bulk RNA from nasal wash samples using the QIAamp Viral RNA Mini Kit (QIAGEN) according to manufacturer's instructions. Where possible, we extracted RNA from 560 μL of sample, the maximum volume recommended for use with the QIAamp kit, to capture as much viral diversity as possible.

To amplify the influenza genome, we modified the primers designed by *Hoffmann et al. (2001)* for full-length amplification of the influenza A genome (*Figure 2—source data 1*). We performed reverse transcription using Superscript III First-Strand Reaction Mix (Thermo Fisher) and an equimolar mix of the 5'-Hoffmann-U12-A4 and 5'-Hoffmann-U12-G4 primers, which bind to the conserved U12 region present on each influenza gene segment. To 6 μL RNA eluent, we added 1 μL annealing buffer and 1 μL of 2 uM primer mix, then incubated at 65 degrees C for 5 min. We added 10 μL 2X First-Strand Reaction Mix and 2 μL Superscript III/RNaseOUT Enzyme Mix on ice for a 20 μL total reaction volume, then incubated at 25 degrees C for 10 min (this initial incubation is designed to help with the binding of short primers), 50 degrees C for 50 min, and 85 degrees C for 5 min.

We used the entire 20 μL volume of the reverse-transcription reaction as template in a 100 μL PCR reaction using KOD HotStart Reaction Mix (EMP Millipore) and a 24-primer cocktail as described in *Figure 2—source data 1* at a total concentration of 600 nM. We performed 35 cycles of PCR amplification with an annealing temperature of 55 degrees C and an extension time of 3 min.

We purified the PCR product using 1X AMPure beads (Beckman Coulter) and prepared libraries for Illumina sequencing using Nextera XT (Illumina). We sequenced the libraries on a NextSeq 500 platform (Illumina) with 150 bp paired-end reads. We performed library preparation and sequencing in duplicate, starting from independent reverse-transcription reactions.

## Read mapping

We first used bowtie2 (*Langmead and Salzberg, 2012*) to filter out reads that mapped to the human genome. Remaining reads are available in the SRA as BioProject PRJNA364676. We used cutadapt 1.8.3 (*Martin, 2011*) to trim adapter sequences from the remaining reads, remove bases at the ends of reads with a Q-score below 25, and filter out reads whose remaining length was shorter than 20 bases. We locally aligned trimmed reads to the A/Brisbane/10/2007 (H3N2) genome (Genbank accessions CY035022 to CY035029) using bowtie2 (*Langmead and Salzberg, 2012*) and tallied the counts of each base at each genome position using custom scripts. We discarded reads with a mapping score below 20, as well as bases with a Q-score below 20.

## Quality filtering

We calculated average sequencing coverage in 50 bp bins along the viral genome. Because we prepared sequencing libraries using Nextera tagmentation, we expect coverage to be low at the two ends of the eight viral gene segments, corresponding to 16 bins. We discarded samples with more than 16 bins with average coverage below 200x (*Figure 2—figure supplement 1A*). We also identified sites at which a non-consensus base reached a frequency of at least 1% in both sequencing replicates and compared variant frequencies between replicates. We discarded samples for which the average difference between variant frequencies in the two replicates exceeded 0.05 (*Figure 2—figure supplement 1B*). In total, we excluded eight samples from downstream analyses. The samples shown in *Figure 1B* are high-quality samples only.

## Variant calling and annotation

For each patient, we identified variable nucleotide sites in the viral genome. We defined these sites as positions with a sequencing coverage of at least 200x, at which multiple bases are present at a frequency of at least 5% in both replicate libraries. We used custom scripts to determine each variant's codon position and whether it created a synonymous or nonsynonymous substitution.

### A note on codon numbering and gene annotation

We numbered HA codons according to the H3 numbering system. This HA numbering scheme assigns 1 to codon 17 of the full HA gene, which is the beginning of the mature HA protein. The codons for all other genes are numbered sequentially beginning with one at the N-terminal methionine. The M1 and M2 genes have 27 bp of in-frame and 44 bp of out-of-frame overlap, and the NS1 and NEP genes have 30 bp of in-frame and 251 bp of out-of-frame overlap. We annotated variants separately for each gene if they occurred in these regions of overlap.

## Phylogenetic analysis

For each patient in our study, we determined the viral consensus sequence at the first sequenced time point. We also downloaded the set of 503 sequences in the Global Initiative on Sharing All Influenza Data (GISAID) EpiFlu database (*Bogner et al., 2006*) corresponding to all full-length HA coding regions from human H3N2 influenza A isolates collected in the USA from January 1, 2004 to December 31, 2007 (GISAID acknowledgement tables provided in *Supplementary file 1*). We analyzed only sequences with passage annotation 'Unpassaged,' 'Original', or 'P0,' indicating that the strains were sequenced directly from the clinical isolates, leaving 63 unique sequences for phylogenetic inference. We pairwise aligned each sequence to the A/Brisbane/10/2007 (H3N2) coding sequence (Genbank accession CY035022) using the program needle from EMBOSS 6.6.0 (*Rice et al., 2000*), which implements a Needleman-Wunsch alignment. We used RAxML 8.2.3 (*Stamatakis, 2014*) to infer a phylogeny from this alignment using a GTRCAT codon-substitution model and visualized the tree using the R package ggtree (*Yu et al., 2017*).

## Haplotype inference

We identified paired-end reads that spanned *n* variable sites of interest within a single gene and determined which bases were present at each variable site. We summarized this information as an *n*-digit binary haplotype, in which each digit represented one variable site, 0 represented the ancestral base, and 1 represented the derived base. We discarded reads that did not span all sites of interest, or that contained genotypes other than the most common derived base. We estimated the rate of PCR recombination as described in *Figure 3—figure supplement 1*. In *Figure 3—figure supplement 2* and *Figure 3—figure supplement 3*, we show the number of paired-end reads used to infer the haplotypes in *Figure 3*.

## Analysis of global variation

To identify sites of global variation in influenza, we downloaded all sequences in the Global Initiative on Sharing All Influenza Data (GISAID) EpiFlu database (*Bogner et al., 2006*) corresponding to all full-length influenza coding regions from human H3N2 influenza A isolates collected from January 1, 2000 to December 31, 2015. Acknowledgement tables are provided as *Supplementary file 1*. We pairwise aligned each sequence to the A/Brisbane/10/2007 (H3N2) coding sequence (Genbank

accession CY035022) using the program needle from EMBOSS 6.6.0 (*Rice et al., 2000*), which implements a Needleman-Wunsch alignment. We calculated the amino-acid distance of each sequence from the Brisbane/2007 reference and excluded outliers whose distance deviated significantly from the other sequences originating from that year, since these sequences may have been misannotated. We tallied the amino acids present at each codon position in each year, discarding sequences that contained indels, and we identified sites at which multiple amino acids were present at a frequency of at least 10% within a single year, or at which the consensus base changed from year to year.

## Statistical tests of parallelism

We sought to test the probability that the parallel emergence of mutations across multiple patients in our study was due to chance. We began with a simple null model in which all sites are equally likely to mutate, and we drew sites from each gene at random without replacement, matching the number of mutations observed in each patient. We calculated the number of unique sites of mutation among all four patients in this simulated data set, and we compared this distribution to the number of unique sites observed in our sequencing data: fewer unique sites of mutation indicates more parallelism (*Figure 2—figure supplement 7A*). This null model is overly simplistic, since some sites in a protein experience more evolutionary constraint. To estimate this constraint, we limited the number of sites considered mutable to the sites that show at least two instances of nonsynonymous mutation in the global H3N2 population between 2000 and 2015 (see Analysis of Global Variation) (*Figure 2—figure supplement 7B*). The p-values given in the main text are calculated under this more conservative null model. We also performed permutation tests for a range of possible proportions of mutable sites and calculated the fraction of simulations that matched or exceeded the amount of parallelism observed in our data (*Figure 2—figure supplement 7C*).

We used a similar approach to test whether the overlap of mutations observed within patients in our study and in the global flu population was likely to be due to chance. We drew two independent sets of sites from each gene at random without replacement, matching the total number of unique variable sites within all patients and the number of variable sites observed in the global population. We then calculated the overlap between these two sets of sites. We used the approach above to calculate the overlap under a simple null model in which all sites in the gene are equally like to mutate; a constrained null model in which the only mutable sites are ones that show nonsynonymous mutation between 2000 and 2015; and across a range of possible constraints (*Figure 4—figure supplement 2*).

## Data and code availability

The FASTQ files are available on the SRA as BioProject PRJNA364676. The computer code that performs the analysis is available at https://github.com/ksxue/parallel-evolution (with a copy archived at https://github.com/elifesciences-publications/parallel-evolution) and in *Supplementary file 2* (*Xue, 2017*).

## Acknowledgements

We thank Choli Lee and Seungsoo Kim for assistance with sequencing; Darneshia Smith for sample management; Louise Kimball and Alpana Waghmare for interpretation of patient clinical data; and Seungsoo Kim, Alexander Greninger, and Trevor Bedford for comments and discussion about the manuscript. We also thank Thomas Friedrich, Louise Moncla, and Nick Florek for helpful discussions about methods for deep-sequencing and Mike Famulare for helpful discussions about evolution at the within- and between-host scales.

## Additional information

### Competing interests

JAE: JAE reports research support from Gilead Sciences, GlaxoSmithKline, Chimerix, and Pfizer, and fees for participation in a Data Safety Monitoring Board for GlaxoSmithKline. MB: MB reports research support and consulting fees from Aviragen Therapeutics, Gilead Sciences, and Ansun

BioPharma, and research support from GlaxoSmithKline. The other authors declare that no competing interests exist.

## Funding

| Funder | Grant reference number | Author |
| --- | --- | --- |
| National Institute of General Medical Sciences | R01GM102198 | Jesse D Bloom |
| National Institute of Allergy and Infectious Diseases | R01AI127893 | Jesse D Bloom |
| National Heart, Lung, and Blood Institute | R01HL081595 | Michael Boeckh |
| Howard Hughes Medical Institute | Faculty Scholar Award | Jesse D Bloom |
| Simons Foundation | Faculty Scholar Award | Jesse D Bloom |
| National Science Foundation | DGE-1256082 | Katherine S Xue |
| Hertz Foundation | Hertz Graduate Fellowship | Katherine S Xue |
| National Heart, Lung, and Blood Institute | K24HL093294 | Michael Boeckh |
| National Heart, Lung, and Blood Institute | K23HL091059 | Angela P Campbell |

The funders had no role in study design, data collection and interpretation, or the decision to submit the work for publication.

## Author contributions

KSX, Conceptualization, Software, Investigation, Visualization, Methodology, Writing—original draft, Writing—review and editing; TS-A, Resources, Data curation, Writing—review and editing; APC, Resources, Data curation, Project administration, Writing—review and editing; JAE, Resources, Data curation, Funding acquisition, Project administration, Writing—review and editing; SAP, Conceptualization, Resources, Data curation, Project administration, Writing—review and editing; MB, Resources, Data curation, Funding acquisition, Project administration; JDB, Conceptualization, Supervision, Funding acquisition, Writing—original draft, Writing—review and editing

## Author ORCIDs

Katherine S Xue, http://orcid.org/0000-0002-4094-3615
Jesse D Bloom, http://orcid.org/0000-0003-1267-3408

## Ethics

Human subjects: Samples were prospectively collected during a surveillance study for respiratory viruses performed in allogeneic hematopoietic stem cell transplant (HCT) recipients undergoing transplantation between December 2005 and February 2010 at Fred Hutchinson Cancer Research Center (Campbell et al. 2015). Following written informed consent, weekly nasal wash samples (or nasopharyngeal swabs if nasal wash samples were precluded clinically) and oropharyngeal swab specimens were obtained at least once before and weekly after HCT up to 100 days. Afterwards, samples were collected as long as the patients continued to test positive for respiratory viruses, if they developed new symptoms, or at least every three months until one year post-transplantation.

# Additional files

### Supplementary files

• Supplementary file 1. GISAID acknowledgement tables for global influenza sequences.

• Supplementary file 2. Code and source data files for all analyses.

## Major datasets

The following dataset was generated:

| Author(s) | Year | Dataset title | Dataset URL | Database, license, and accessibility information |
| --- | --- | --- | --- | --- |
| Xue KS, Bloom JD | 2017 | Deep sequencing data | http://www.ncbi.nlm.nih.gov/bioproject/PRJNA364676 | Publicly available at the NCBI BioProject database (accession no: PRJNA364676) |

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
