## [Decision Letter]

Thank you for submitting your article "Parallel evolution of influenza across multiple spatiotemporal scales" for consideration by *eLife*. Your article has been favorably evaluated by Patricia Wittkopp (Senior Editor) and two reviewers, one of whom, Richard A Neher (Reviewer #1), is a member of our Board of Reviewing Editors.

The reviewers have discussed the reviews with one another and the Reviewing Editor has drafted this decision to help you prepare a revised submission.

Summary:

The reviewers agreed that you present a high-quality data set that quantifies and tracks within host evolution of influenza A virus at unprecedented detail. Similar variants are observed in viral populations in different immuno-compromised individuals and the variable sites have unexpected overlap with polymorphic sites in the global epidemic. This raises the possibility that mutation observed within host can inform predictions of influenza strains dominating future seasons. These are important observations.

Essential revisions:

However, before we can recommend your manuscript for publication in *eLife*, we would like you to address the following points.

1) Be careful and explicit about what can be concluded from the observed parallelism of mutations. The statement "Influenza evolution within single infected hosts acts as a microcosm of viral evolutionary dynamics observed at the global scale" overstates the degree to which global influenza evolution is predicted by within host patterns. In our interpretation, your data suggest that mutations observed within host in HA are enriched for mutations that are polymorphic later in the global epidemic. Beyond that, quite some uncertainty remains. Fixation within host does not imply fixation globally (site 138), and a site observed transiently within host (site 223) might fix globally. The statement that "most" recurrent mutations have the same within host/global dynamics (subsection “Within-host evolution often parallels global evolutionary dynamics”, last paragraph) is not supported by your data and this needs to be toned down. The degree of parallelism is remarkable, but you should be careful not to overstate it.

2) You use sequences from GISAID. Hence you need to provide a table acknowledging the GISAID contributors (or use only public domain sequences).

3) Thank you for providing the analysis code. It would be useful to if you could check this code into a git repo (e.g. on github) that *eLife* can then clone. A top-level README that explains the content of the different folders and its relation to the paper would also help.

4) Please extend the discussion of the statistical tests for parallelism in the main text (subsection “Within-host evolution often parallels global evolutionary dynamics”, first paragraph) to give the reader a better idea of what is being tested. Please also discuss the fact that significant enrichment is only observed when at least half of all amino acids in HA are assumed mutable.

---

## [Author Response]

*Essential revisions:*

*However, before we can recommend your manuscript for publication in eLife, we would like you to address the following points.*

*1) Be careful and explicit about what can be concluded from the observed parallelism of mutations. The statement "Influenza evolution within single infected hosts acts as a microcosm of viral evolutionary dynamics observed at the global scale" overstates the degree to which global influenza evolution is predicted by within host patterns. In our interpretation, your data suggest that mutations observed within host in HA are enriched for mutations that are polymorphic later in the global epidemic. Beyond that, quite some uncertainty remains. Fixation within host does not imply fixation globally (site 138), and a site observed transiently within host (site 223) might fix globally. The statement that "most" recurrent mutations have the same within host/global dynamics (subsection “Within-host evolution often parallels global evolutionary dynamics”, last paragraph) is not supported by your data and this needs to be toned down. The degree of parallelism is remarkable, but you should be careful not to overstate it.*

We appreciate the suggestions to add greater specificity to our claims, and we have made several textual revisions in response. We have revised our original impact statement from "Influenza evolution within single infected hosts acts as a microcosm of viral evolutionary dynamics observed at the global scale” to “Influenza evolution within infected hosts recapitulates many evolutionary dynamics observed at the global scale,” a more specific statement from our Abstract.

In the Results section, we have changed one of the section titles to state that “Within-host variants frequently arise at sites that are polymorphic in influenza globally.” We have also removed the statement that “within-host evolution can sometimes be a microcosm of global evolution” in favor of more specific language about the fixation (or lack thereof) of the specific mutations that the reviewers point out. We also specifically highlight certain differences at the within- and between-host scales like the mutation Y193, which arises at a globally variable site but is never itself observed on at a global scale.

*2) You use sequences from GISAID. Hence you need to provide a table acknowledging the GISAID contributors (or use only public domain sequences).*

We have added tables acknowledging the GISAID contributors to the Github repository for this manuscript: https://github.com/ksxue/parallel-evolution/tree/master/acknowledgements We have also included these tables as [Supplementary-material SD3-data] of the revised manuscript. Because we use tens of thousands of sequences in our analyses, the tables are divided into smaller increments that could be downloaded from the GISAID server.

*3) Thank you for providing the analysis code. It would be useful to if you could check this code into a git repo (e.g. on github) that eLife can then clone. A top-level README that explains the content of the different folders and its relation to the paper would also help.*

We appreciate the opportunity to make our analysis code more widely available. We have deposited the code within the Github repository here: https://github.com/ksxue/parallel-evolution Per the reviewer suggestions, we have also added a top-level README file that describes the overall organization of the repository. The “analysis” folder, which contains most of the code used to generate the figures, also includes individual README files for each discrete analysis.

*4) Please extend the discussion of the statistical tests for parallelism in the main text (subsection “Within-host evolution often parallels global evolutionary dynamics”, first paragraph) to give the reader a better idea of what is being tested. Please also discuss the fact that significant enrichment is only observed when at least half of all amino acids in HA are assumed mutable.*

We substantially expanded our description of the statistical tests for parallelism in the main text, which we believe will benefit the readers.